# Characteristics of Emergencies in the Workplace from the Perspective of the Emergency Medical Services: A 4-Year Case-Control Study

**DOI:** 10.3390/ijerph20031863

**Published:** 2023-01-19

**Authors:** Krzysztof Marek Mitura, Daniel Celiński, Paweł Jastrzębski, Piotr Konrad Leszczyński, Robert Gałązkowski, Sławomir Dariusz Szajda

**Affiliations:** 1Independent Public Health Care Center RM-MEDITRANS Emergency Station and Sanitary Transport in Siedlce, 08-110 Siedlce, Poland; 2Department of Emergency Medical Service, Medical University of Warsaw, 02-091 Warsaw, Poland; 3Department of Emergency Medical Service, Faculty of Health Sciences Collegium Medicum, University of Warmia and Mazury, 10-719 Olsztyn, Poland; 4Faculty of Medical Sciences and Health Sciences, Siedlce University of Natural Sciences and Humanities, 08-110 Siedlce, Poland

**Keywords:** medical rescue, illness, accident, workplace, employee, patient

## Abstract

Introduction: Accidents and emergencies in the workplace account for a significant proportion of emergency calls worldwide. The specificity of these events is often associated with hazards at a given workplace. Patients do not always require hospitalization; therefore, the characteristics of events can only be determined from the perspective of emergency medical services teams. The aim of the study was to analyze calls and the course of emergency ambulance interventions to patients at their workplace. Material and methods: The study was conducted based on a retrospective analysis of data contained in the medical records of the ambulance service from central Poland from 2015–2018. From all interventions (*n* = 155,993), 1601 calls to work were selected, and the urgency code, time of day and year, patients’ sex, general condition, as well as diagnoses according to the International Classification of Diseases—ICD-10 and the method of ending the call were considered. Results: The mean age of patients in the study group was 42.4 years (SD ± 13.5). The majority were men (*n* = 918; 57.3%). The number of calls increased in the autumn (*n* = 457; 28.5%) and in the morning (*n* = 609; 38.0%). The main reasons for the intervention were illnesses (ICD-10 group: R—‘symptoms’) and injuries (ICD-10 group: S, T—‘injuries’). Calls at workplaces most often ended with the patient being transported to the hospital (78.8%), and least often with his death (0.8%). Conclusions: The patient profile in the workplace indicates middle-aged men who fall ill in the fall, requiring transport to the hospital and further diagnostics.

## 1. Introduction

Due to professional activity, human life can be divided into three periods: pre-working age, working age and post-working age. There are a number of working age limits, and the framework may differ depending on gender. The Organization for Economic Cooperation and Development (OECD) defines the working-age population as people aged 15 to 64 [1].

In Poland, the working age is defined by the Central Statistical Office [2,3] as the age of ability to work, starting at the age of 18 and ending at the age of 60 for women and 65 for men. In 2020, the working age population of Poland accounted for 59.5% (*n* = 22,771.4), including women 27.9% (*n* = 10,683.7 thousand) and men 37.6% (*n* = 12,087.7) [3]. 

People of working age directly influence the economic development of the country through their professional activity. Thus, each country should take special care of people of working age. This concern should begin with extensive health prevention programs and end with the provision of high-quality medical assistance, especially in situations of a sudden threat to health and life. Such a systemic approach to the health of professionally active people allows for early treatment of the disease, reducing the number of days of incapacity for work and possible disability benefits, which translates into budget savings.

All this means that one of the basic tasks of the state is to provide assistance to people who are in a state of sudden health risk. Such a state should be understood as the occurrence of sudden or expected appearance of symptoms of deterioration of health, the consequence of which may be serious damage to the body’s functions, bodily injury or loss of life [4]. For this purpose, the State Medical Rescue (SMR) system was created in Poland. The key element of SMR is emergency medical services (EMS teams, deployed in such a way as to ensure the proper parameters of reaching the event site [4]).

Work-related injuries, illnesses, and deaths result in high social and economic costs for workers and their families as well as for employers and governments [5,6]. These costs may include damage to health resulting in early retirement, disability pensions, compensation, shortages of qualified employees from the labor market, and absenteeism from work, as well as high costs of treatment and insurance premiums [5].

In 2020, 191.187 million people aged 15–64 worked in the European Union (EU), including 16.049 million in Poland [7]. Out of this number, in 2020, 4.7% (in the EU) and 12.8% (in Poland) of employees aged 15–64 had health problems related to their work, with health problems resulting in at least four days of absence from work. Additionally, 3.3% of employees in the EU and 8.4% of employees in Poland, or at least one month’s absence, was reported by 1.8% of employees in the EU and 2.8% in Poland [8]. In 2019, 2,377,000 accidents at work were recorded in the European Union, and 61,233 in Poland, of which 2,374,000 in the EU and 61,084 in Poland involved absence from work for at least four days, and 3008 fatal accidents were recorded, with 149 of them from Poland [9].

In Poland, economic burdens resulting from work-related injuries and diseases are relatively high and their average value is 10.2% (median 7.2%) of gross domestic product—a measure of economic development [5]. 

Review of the literature: It should be stated that currently there are no studies on the intervention of emergency medical service teams in workplaces. This work is an attempt to fill this gap, and is a contribution to further in-depth analyses on the above issues.

Aim: The main purpose of the work is to analyze EMS team interventions in workplaces in central Poland in 2015–2018. Specific objectives included an analysis of EMS team activities based on the urgency code and the average time to reach the scene, patient’s sex and age, seasons, months, days of the week, call times, the most frequent diagnoses and groups of diagnoses, and ways of completing the intervention.

## 2. Materials and Methods

The study area covered an area of 7350 km^2^ and at the end of 2018 it was inhabited by 547,866 people (women *n* = 270,222; 49.3%, men *n* = 277,644; 50.7%), with people of working age accounting for 60.03% (*n* = 328,876), of which women *n* = 150,924 (55.9%—of the entire female population) and men *n* = 177,952 (64.1%—of the entire male population) [10]. In addition, 69.4% of the inhabitants of the study area live in rural areas and 30.6% in urban areas with urban areas accounting for only 1.6% of the study area. [10]. 

The study was conducted on the basis of a retrospective analysis of data contained in the medical records kept by EMS in the years: 2015 (*n* = 36,687), 2016 (*n* = 37,701), 2017 (*n* = 39,689), and 2018 (*n* = 41,916). 

Out of the analyzed 155,993 EMS orders from 2015–2018, 1725 cases (1.1%) were defined as workplaces by the medical dispatcher or EMS manager, where there was no patient in three interventions, one was false, one was cancelled during the arrival to the place of call, and in 111 the head of the EMS did not specify the patient’s age. Among the 111 (women *n* = 43, men *n* = 68) interventions in which the patient’s age was not specified, the following diagnoses prevailed: *R55-Syncope and collapse n* = 28 (women *n* = 17, men *n* = 11), *G40-Epilepsy n* = 9 (female *n* = 1, male *n* = 8), *S01-Open wound of head n* = 9 (female *n* = 1, male *n* = 8), and *R07-Pain in the throat and the chest n* = 5 (women *n* = 2, men *n* = 3).

Excluding the false ones from all call outs (*n* = 8136), where the patient was not at the place of the call, the patient refused to help, or when the patient’s age or gender was not specified in the documentation, 147,857 interventions were selected (women *n* = 69,740; 47.2%, men *n* = 78117; 52.8%). In this group of interventions, calls to workplaces accounted for *n* = 1601; 1.1% (women *n* = 683, men *n* = 918), and the number of EMS interventions to workplaces in individual years was: *n* = 281 (2015), *n* = 356 (2016), *n* = 448 (2017), and *n* = 516 (2018).

Individual data from individual EMS team interventions in workplaces are presented quantitatively depending on the urgency code, gender, the patient’s age, season, month, day of the week, hours of the call, and the 10 most common groups of diagnoses according to the International Statistical Classification of Diseases and Health Problems (ICD-10), the 10 most common diagnoses made based on the ICD-10 codification, and the method of completing the intervention by the EMS team. The urgency codes are defined as follows:


*Code 1—EMS team departure time from being dispatched by a medical dispatcher is a maximum of 1 minute; EMS team absolutely use light and sound signals, broadcast in the event of sudden cardiac arrest, loss of consciousness, stroke, traffic accidents, etc.*



*Code 2—EMS team departure time from being dispatched is a maximum of 2 minutes; the head of the EMS decided on the use of light and sound signaling, from 2019 the EMS team reaction time is 3 minutes, the medical dispatcher decides on the use of signaling, broadcast in the case of mental disorders, injuries, etc.*


The data was collected in the Microsoft Excel MS Office 2021 database for Windows11. The obtained results were statistically analyzed using the STATISTICA 13.3 program by TIBCO. In the statistical analysis, the Chi-square test of independence, Pearson’s correlation, was used to check the relationship between qualitative features. The data are presented as numbers (*n*) and percentages (%). Statistically significant results were considered at *p* < 0.05. 

## 3. Results

In the analyzed period (2015–2018), out of the total number of EMS team instructions (*n* = 155,993), the most frequent calls were reported to the patient’s home (*n* = 119,561; 76.56%), public places (*n* = 17,422; 11.2%), in road and road traffic (*n* = 8134; 5.2%), clinics (*n* = 2608; 1.7%), schools (*n* = 2265; 1.5%), nursing homes (*n* = 1869; 1.2%), workplaces (*n* = 1725; 1.1%) and other call places (*n* = 2409; 1.5%). 

In the analyzed period we found a gradual increase in the number of EMS interventions in workplaces (*n* = 281 in 2015, *n* = 356 in 2016, *n* = 448 in 2017, *n* = 516 in 2018), and this increase between 2015 and 2018 amounted to 83.6%. (*n* = 235).

The conducted research shows that EMS teams were more often dispatched to workplaces in the urgency code 2 (57.5%) than in the urgency code 1 (42.5%). The highest percentages of EMS team interventions in men were in code 1 (70.3%), and in women in code 2 of urgency (52.3%). The analysis of EMS team interventions at workplaces shows significant differences (χ2 = 81.9, *p* < 0.001) depending on the urgency code, considering gender (Figure 1, Table 1). 

In workplaces, EMS teams intervened more often with men (57.3%) than with women (42.7%). The highest percentages of EMS team trips for women were recorded in the age groups of 70 and more (69.0%) and 50–59 (48.8%), and for men in the age group of 60–69 (75.6%) and 20–29 years (60.7%). Statistical analysis suggests statistically significant differences (χ2 = 42. 8, *p* < 0.001), depending on the patient’s age and sex (Figure 2, Table 2).

EMS team interventions to workplaces by gender and season (χ2 = 6.7, *p* = 0.1), month (χ2 = 17.4, *p* = 0.1), day of the week (χ2 = 7.5, *p* = 0.3), and hourly intervals (χ2 = 10.2, *p* = 0.1) show no significant relationships. 

EMS teams most often left for workplaces in autumn (*n* = 457; 28.5%). In turn, the month of the most frequent interventions was September (*n* = 156; 9.7%) with (Figure 3), Fridays (*n* = 303; 18.9%) being the most frequent, and the hourly ranges: 08:01–12:00 (*n* = 609; 38.0%). EMS teams were called to workplaces least frequently in winter (*n* = 299; 18.7%), in the month of February (*n* = 88; 5.5%) (Figure 3), on Sunday (*n* = 45; 2.8%), during hours: 00:01–04:00 (*n* = 52; 3.3%) and 20:01–24:00. In the case of interventions for women, they most often took place in autumn (*n* = 210; 30.8%), September (*n* = 74; 10.8%) and October (*n* = 72; 10.5%) (Figure 3), on Fridays (*n* = 141; 20.6%) and Tuesdays (*n* = 134; 19.6%), during 08:01–12:00 (*n* = 268; 39.2%), and least frequently in winter (*n* = 137; 20.1%), February (*n* = 40; 5.9%) and July (*n* = 44; 6.44), respectively %) (Figure 3), Sunday (*n* = 19; 2.8%), during 00:01–04:00 (*n* = 18; 2.6%). In turn, EMS team interventions in men’s workplaces were most common in the summer (*n* = 261; 28.4%), in June (*n* = 92; 10.0%) (Figure 3), on Monday (*n* = 179; 19.5%), during hours: 08:01–12:00 (*n* = 341; 37.2 %). The least frequent interventions were in winter (*n* = 162; 17.7%), February (*n* = 48; 5.2%) (Figure 3), and Sunday (*n* = 26; 2.8%) during 00:01–04:00 (*n* = 34; 3.7%). 

Of the 10 most common groups of ICD-10 diagnoses according to which EMS teams make diagnoses, EMS managers most often based their diagnosis on the group *R-Symptoms, signs, and abnormal clinical and laboratory findings, not elsewhere classified* (40.9%), and least often *Z-Factors influencing health status and contact with health services* (1.1%). With regard to women, the highest percentages were obtained in the case of diagnosis groups *F-Mental and behavioral disorders* (61.1%) and *R-Symptoms, signs, and abnormal results of clinical examinations not elsewhere classified* (57.7%), and in the case of men *V-Y-External causes of morbidity and mortality* (86.36%) and *G-Diseases of the nervous system* (75.7%). The analysis of the obtained results indicates statistically significant differences (χ2 = 165.4, *p* < 0.001) between the groups of ICD-10 diagnoses depending on the sex of the patients who were provided medical assistance at the workplace by EMS teams (Table 3).

The most common diagnoses in workplace interventions were diagnoses *R55- Syncope and collapse* (19.2%) and *R07-Pain in throat and chest* (6.1%). The highest percentage of interventions in women was found when the diagnosis was made *R51-Headache* (78.8%) and *R10-Abdominal and pelvic pain* (68.3%), and in men *R56-Convulsions, not elsewhere classified* (97.6%) and *G40-Epilepsy* (86.4%). Statistically significant differences were found between the sex of the patients of the diagnoses made by EMS managers based on the ICD-10 codification, taking into account (χ2 = 189.4, *p* < 0.001) (Table 4).

In the analyzed period, 10 interventions were recorded in workplaces in which the team leader made a diagnosis of *I21-Acute myocardial infarction*, two women aged 57 and 62, and eight men aged 47 to 71. All of them were transported from the scene directly to the hemodynamics laboratory. There were also 11 diagnoses of *I64-Stroke, not specified as hemorrhage or infarction* in five women aged 30–59 and six men aged 43–65. Finally, in the case of a 45-year-old man, the actions taken turned out to be ineffective, while five patients aged 54–67 were transported to the hospital. In nine cases, alcohol consumption was the main reason for EMS interventions at workplaces. In three cases, the diagnosis *F10 -Mental and behavioral* was made, in a woman aged 30 and in two men aged 46 and 56. In one case *T51 -Toxic effect of alcohol* was diagnosed (a man aged 44) and in five cases *Y91-Evidence of alcohol involvement determined by level of intoxication*, men aged 21–52. 

EMS calls at workplaces most often ended with the patient being transported to the hospital (78.8%), and least often with his death (0.8%). In the case of women, the highest percentage was recorded when, after the EMS team intervention, the patients remained at the place of call (50.3%), and in the case of men, when they were handed over to the air ambulance team (100.0%) or the patient died (100,0%). Statistically significant differences were found for EMS team calls to workplaces in relation to the method of completing the intervention, considering the patient’s sex (χ2 = 31.5, *p* < 0.001) (Table 5).

Among those transported to hospitals, six patients were admitted to psychiatric hospitals (two women aged 48 and four men aged 24–48). In workplaces, there were 13 cases of abandonment of emergency medical activities by EMS teams due to the patient’s death (men aged 41–81), including one case of suicide of a man aged 46 (*X70-Intentional self-injury by hanging, strangulation, and choking, Intentional self- harm by hanging, strangulation, and suffocation).*

## 4. Discussion

Based on our own research, we found that EMS teams stationed in Central-Eastern Poland were dispatched to workplaces in 1.1% of all their interventions, with EMS teams more often providing medical assistance to men (57.3%) than women (42.7%). 

On the scale of Poland in 2015–2018, out of all EMS interventions, instructions to people aged 18–64 accounted for 50.5% (2018) to 51.9% (2015). In the case of people of pre-working age (less than 18 years of age), interventions ranged from 6.0% (2018) to 6.4% (2015), while callouts to people aged 65 and more accounted for 41.7% (2015) to 45.4% (2018) [11]. In the same period, in the Mazowieckie Voivodship (Poland), EMS visits to people in pre-working age ranged from 5.5% (2017) to 6.2% (2016), interventions in patients aged 18–64 from 48.5 % (2018) to 52.3% (2015), and in the case of people aged 65 and more, they accounted for from 41.2% (2015) to 45.8% (2018) of all EMS callouts [11]. However, according to Białczak et al. [12], in the northern part of the Mazowieckie Voivodeship (Poland), in 2013–2016, EMS interventions for people in pre-working age amounted to 5.9%, for those aged 18–59, it was 43.3%, and for 60 years and older it was 50.8%.

In our own study, as in the studies of other authors, an increase in the number of EMS interventions at workplaces was found. In 2015–2018, callouts of emergency medical services teams in Poland to the place of an event defined as work ranged from 1.8% (2016) to 2.1% (2018) of all dispositions. In the case of the Mazowieckie Voivodship, the range was from 1.8% (2016) to 2.4% (2018) [10]. Filip et al. [13] report that in 2012, in the Provincial Emergency Services in Rzeszów (Poland), interventions to workplaces accounted for 1.4% of all EMS trips. On the other hand, Pittet et al. [14] indicates that in the period from 2001 to 2010 in the canton of Vaud (Switzerland), interventions to workplaces and schools ranged from 3.6% (2004) to 4.3% (2007). 

Aftyka and Rudnicka-Drożak [15], based on data from the Provincial Emergency Services in Lublin (Poland), regarding all dispositions, regardless of the patient’s age and place of the event, indicated a higher percentage of EMS interventions in women (51.1%) than in men (48.9%), while Pittet et al. found that about half of the EMS instructions concerned men (range from 48.5% to 50.2%) [14].

Based on our own analyses, we found that EMS teams to workplaces were more often available in code 2 (57.5%) than in code 1 (42.54%). Irrespective of the place of the event and the patient’s age, in the years 2013–2017 in the district of Mińsk (Poland), EMS instructions in the urgency code 1 accounted for 29.0%, and in the code 2 for 71.0% [16]. On the other hand, Filip et al. [13] report that in code 1 EMS services were performed in 90.6%, and in code 2 in 9.4% of all interventions. On the other hand, Andrzejewski M. et al. [17] report that in the Łódź Region (Poland) of the Falck substation, between 1 January and 30 April 2015, emergency medical service team callouts in code 1 accounted for 67%, and in code 2 they accounted for 33% of interventions. Conversely, Białczak et al. [12] showed that EMS callouts in code 1 accounted for 34.4%, and in code 2 for 64.1% of all interventions. The authors suggest increasing security controls or changing the organization of workplaces on the days (and selected times) when the highest number of incidents is recorded.

Based on our own research, we found that EMS callouts to workplaces in relation to gender, season, month, day of the week, and hourly intervals of interventions do not show statistically significant differences. Our analyses show that EMS teams most often left for work in autumn (28.5%), in September (9.7%), and on Fridays (18.9%) between 08:01–12:00 (38.0%), and least often in winter (18.7%), in February (5.5%), and on Sundays (2.8%) between 00:01–04:00 (*n* = 49; 3.3%). However, regardless of the place of intervention, sex, and age of the patient, Andrzejewski et al. [17] point to the highest number of EMS instructions between 12:00 and 18:00, the least frequent at night, the highest number of calls in January, and the lowest in April. In the northern part of the Mazowieckie Voivodeship (Poland), in 2013–2016, EMS teams most often intervened on Saturdays (15.0% of all calls) and Sundays (15,0%), and the least often on Tuesdays (13.5%) [12]. In the Rzeszów ambulance service in 2012, the most calls were made between 8:00 and 11:59, and the fewest between 00:00 and 3:59 [13]. On the other hand, Møller et al. [18] report that in Copenhagen (Denmark) between December 2011 and November 2013, the highest number of emergency calls regarding the possibility of life and health threats were made in winter (25.5% of all calls), on Saturdays (15, 7%), and during the day (38.9%).

Through our research, we found that in workplaces, EMS interventions concerned calls whose causes were most often included in the ICD-10 codification from the *R-Symptoms, signs, and abnormal clinical and laboratory findings, not elsewhere classified* (40.9%), *S-T-Injury, poisoning, and certain other consequences of external causes* (32.4%), *I-Diseases of the circulatory system* (7.9%), and *G-Diseases of the nervous system* (6.9%). In the case of women, these were most often diagnoses from groups *R, S-T, I, G*, and men *S-T, R, G, I*. The most common diagnoses based on ICD-10 codifications are *R55-Syncope and collapse* (19.2%), *R07-Pain in throat and chest* (6.1%), *G40-Epilepsy* (5.0%), and *S01-Open wound of head* (3.8%); for women *R55, R07, R10-Abdominal and pelvic pain*, *I10-Essential (primary) hypertension* and for men *R55, G40, S01* and *R07*.

In accidents at work in 2019 in the EU, the most common were wounds and superficial injuries (32.0%), dislocations, sprains, sprains and tears (24.5%), internal injuries (19.0%), and bone fractures (11.1%).; in Poland it was, respectively, 48.5%, 22.8%, 3.9%, and 17.8% [9]. In 2017, the most common accidents at work in Poland included limb injuries 77.9% and head injuries 9.8%% [19].

According to Szpakowski and Pilip [20], in the years 2013–2015 in central-eastern Poland, the main reason for calling EMS, regardless of the place of the call and the patient’s age, were diseases of the circulatory and respiratory systems (78%). Mitura et al. [16], in a study on EMS calls in the district of Mińsk, state that interventions are dominated by ICD-10 code groups *R* (37.4%), *S-T* (20.6%), and *I* (15.6%). On the other hand, Białczak et al. [12] indicate, among others, the main reasons for EMS intervention, among others, cardiovascular diseases (36.1%) and injuries (20.1%). On the other hand, Cantwell et al. [21], based on the data of the ambulance service in Melbourne (Australia) from 2008–2011, based on the ICD-10 codification, indicate the most common reasons for EMS interventions are circulatory system disorders (15.6%), injuries, and poisoning (13.5%) as well as diseases of the nervous system (10.4%). The most common diagnoses made by EMS teams based on the ICD-10 codification in Northern Denmark in 2007–2014 are *S-T-Injury, poisoning, and certain other consequences of external causes* (in the range of 26.3% to 34.0%), *R-Symptoms, signs, and abnormal clinical and laboratory findings, not elsewhere classified* (from 14.7% to 28.0%), *Z-Factors influencing health status and contact with health services* (9.6% to 16.5%) and *I-Diseases of the circulatory system* (9.5–11.5%) [22]. In turn, Pittet et al. [14] indicate injuries, comas, and chest pain as the main reasons for EMS intervention in the Swiss canton of Vaud in 2001–2010.

Filip et al. [14] mention circulatory system disorders 26.0% (*I44-I19, R07*), hypertension 15.7% (*I10*), abdominal pain 15.1% (*R10*), and fainting 12.3% (*R55*) as the most common reasons for EMS interventions. The research by Mitura et al. [16] shows that the main diagnoses made by EMS teams, regardless of the place of call and the patient’s age, were *R55-Syncope and collapse* (7.7%) and *R07-Pain in the throat and chest* (5.6%). 

The distribution of diagnoses in the case of working-age patients who were intervened by EMS teams at the workplace shows differences in relation to the distribution of diagnoses made by EMS teams, regardless of the place of call and the patient’s age. [23,24] These differences may result from the types of injuries and diseases depending on the patient’s age and the place where EMS teams are called to. It should be noted that the diagnoses made in pre-hospital care are often not very precise and, as shown, are often based on the ICD-10 codification from the *R-Symptoms, signs, and abnormal clinical and laboratory findings, not elsewhere classified*. This group includes a number of diagnoses concerning various systems; moreover, it includes ambiguous symptoms and disease states, thus these diagnoses are willingly used by EMS teams [25,26].

Based on our own research, we found that after the intervention of working-age patients, EMS teams most often transported them to hospitals (78.8%), and in 19.2% of cases, patients remained at the place of call. In the case of EMS teams in Minsk, Mitura et al. [16] report that 68.6% of patients (from all interventions) were transported to hospitals, and 24.0% remained at the call site. In turn, Filip et al. [13] noted in the period under study that EMS teams transported patients to hospitals in 68.6% of calls, and in 18.8% of interventions, assistance was provided at the call site. On the other hand, the analyses of Andrzejewski et al. [17] show that from all interventions, EMS teams transported 27.8% of patients to the HED, 33.9% to the IP, and 38.4% remained at the call site.

Study limitations: Based on the available data, it was not possible to indicate the type of workplace in which the EMS team intervened, or the profession of the patient who was provided with medical assistance. The study does not verify pre-hospital diagnoses with hospital-confirmed diagnoses and is based on a qualitative data.

## 5. Conclusions

An increase in the number of emergency medical teams’ interventions to workplaces was observed. EMS in workplaces more often intervened to men than to women. The more frequent reason for the intervention of emergency medical teams at workplaces was illness than injuries, and the interventions most often ended with patients being transported to the hospital.

It seems advisable to carry out more effective information and prevention activities in workplaces regarding health safety and injury, and to raise awareness of the benefits of preventive examinations, especially in the case of middle-aged men. Particular attention should be paid to the possibility of sudden illnesses and injuries, especially in the autumn, on Fridays and in the morning. Information and preventive activities may limit the number of EMS interventions in workplaces.

## Figures and Tables

**Figure 1 ijerph-20-01863-f001:**
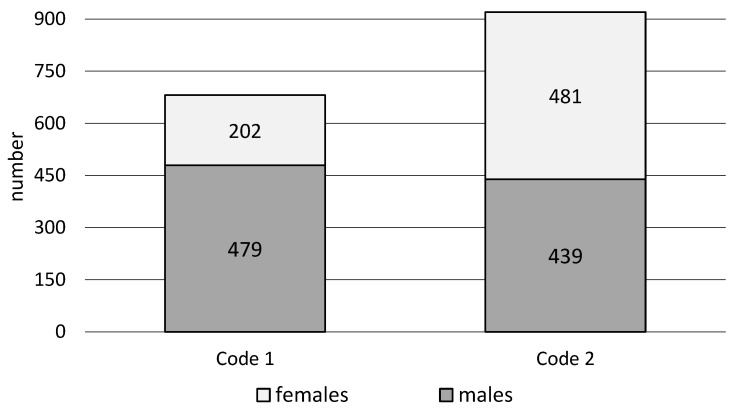
Interventions of emergency medical services teams in the workplace depending on the urgency code and gender.

**Figure 2 ijerph-20-01863-f002:**
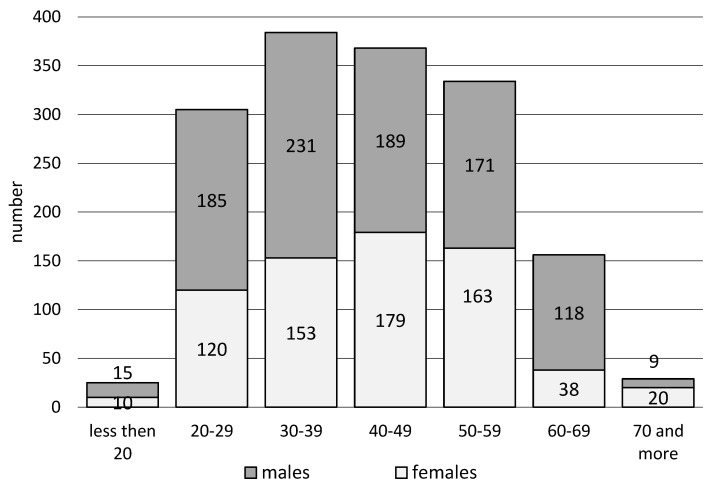
Interventions of emergency medical services teams in the workplace depending on the patient’s age and gender.

**Figure 3 ijerph-20-01863-f003:**
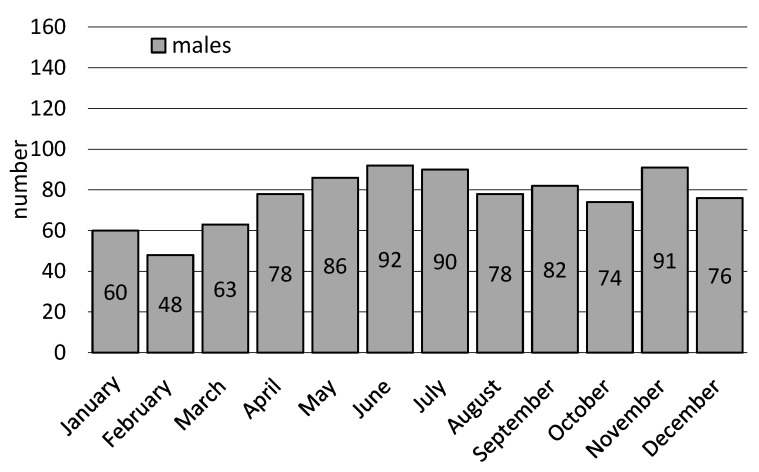
Interventions of emergency medical services teams in the workplace depending on month and the patient’s gender.

**Table 1 ijerph-20-01863-t001:** Interventions of emergency medical services teams in the workplace depending on the urgency code and gender.

Urgency Code	Gender				Total	
	Females		Males			
	*n*	%	*n*	%	*n*	%
Code 1	202	29.7%	479	70.3%	681	42.5%
Code 2	481	52.3%	439	47.7%	920	57.5%
all	683	42.7%	918	57.3%	1601	100.0%

statistical analysis: Pearson χ2 = 81.9 df = 1 *p* < 0.001.

**Table 2 ijerph-20-01863-t002:** Interventions of emergency medical services teams in the workplace considering the patient’s age and gender.

Age	Gender				Total	
	Females		Males			
	*n*	%	*n*	%	*n*	%
<20	10	40.0%	15	60.0%	25	1.6%
20–29	120	39.3%	185	60.7%	305	19.0%
30–39	153	39.8%	231	60.2%	384	24.0%
40–49	179	48.6%	189	51.4%	368	23.0%
50–59	163	48.8%	171	51.2%	334	20.9%
60–69	38	24.4%	118	75.6%	156	9.7%
>70	20	69.0%	9	31.0%	29	1.8%
all	683	42.7%	918	57.3%	1601	100.0%

statistical analysis: Pearson χ2 = 42.8 df = 6 *p* < 0.001.

**Table 3 ijerph-20-01863-t003:** ICD-10 diagnosis groups used by emergency medical services teams to make diagnoses during workplace interventions depending on gender.

Gender						Total		
Females			Males					
Group of ICD-10	*n*	%	Group of ICD-10	*n*	%	Group of ICD-10	*n*	%
**R**	377	57.7%	**R**	277	42.4%	**R**	654	40.9%
**S-T**	134	25.8%	**S-T**	385	74.2%	**S-T**	519	32.4%
**I**	66	52.4%	**I**	60	47.6%	**I**	126	7.9%
**G**	27	24.3%	**G**	84	75.7%	**G**	111	6.9%
**V–Y**	6	13.6%	**V–Y**	38	86.4%	**V–Y**	44	2.8%
**F**	22	61.1%	**F**	14	38.9%	**F**	36	2.3%
**M**	18	50.0%	**M**	18	50.0%	**M**	36	2.3%
**N**	11	52.4%	**N**	10	47.6%	**N**	21	1.3%
**E**	5	26.3%	**E**	14	73.7%	**E**	19	1.2%
**Z**	7	41.2%	**Z**	10	58.8%	**Z**	17	1.1%
**other**	10	55.6%	**other**	8	44.4%	**other**	18	1.1%
**all**	683	42.7%	**all**	918	57.3%	**all**	1601	100.0%

statistical analysis: Pearson χ2 = 165.4 df = 10 *p* < 0.001; **E** = Endocrine, nutritional and metabolic diseases; **F** = Mental and behavioral disorders; **G** = Diseases of the nervous system; **I** = Diseases of the circulatory system; **M** = Diseases of the musculoskeletal system and connective tissue; **N** = Diseases of the genitourinary system; **R** = Symptoms, signs, and abnormal clinical and laboratory findings, not elsewhere classified; **S–T** = Injury, poisoning, and certain other consequences of external causes; **V–Y** = External causes of morbidity and mortality; **Z** = Factors influencing health status and contact with health services.

**Table 4 ijerph-20-01863-t004:** Top 10 ICD-10 diagnoses used by emergency medical services teams during workplace interventions by gender.

ICD-10	Gender				Total	
	Females		Males			
	*n*	%	*n*	%	*n*	%
**R55**	198	64.5%	109	35.5%	307	19.2%
**R07**	52	53.6%	45	46.4%	97	6.1%
**G40**	11	13.6%	70	86.4%	81	5.1%
**S01**	15	24.6%	46	75.4%	61	3.8%
**R10**	41	68.3%	19	31.7%	60	3.8%
**I10**	35	58.3%	25	41.7%	60	3.8%
**S00**	19	33.3%	38	66.7%	57	3.6%
**S61**	11	25.0%	33	75.0%	44	2.8%
**R56**	1	2.4%	41	97.6%	42	2.6%
**R51**	26	78.8%	7	21.2%	33	2.1%
**other**	274	36.1%	485	63.9%	759	47.4%
**all**	683	42.7%	918	57.3%	1601	100.0%

statistical analysis: Pearson χ2 = 189.4 df = 10 *p* < 0.001; **G40** = Epilepsy; I10 = Essential (primary) hypertension; **R07** = Pain in throat and chest; **R10** = Abdominal and pelvic pain; **R42** = Dizziness and giddiness; **R51** = Headache; **R55** = Syncope and collapse; **R56** = Convulsions, not elsewhere classified; **S00** = Superficial injury of head; **S01** = Open wound of head; **S09** = Other and unspecified injuries of head; **S061** = Open wound of wrist and hand.

**Table 5 ijerph-20-01863-t005:** Conclusions of interventions by emergency medical services teams in the workplace by gender.

Patient	Gender				Total	
	Females		Males			
	*n*	%	*n*	%	*n*	%
Left at the place of call	155	50.3%	153	49.7%	308	19.2%
Transport to hospital	528	41.8%	733	58.1%	1261	78.8%
Transferred to Helicopter Emergency Medical Service	0	0.0%	19	100.0%	19	1.2%
Death	0	0.0%	13	100.0%	13	0.8%
all	683	42.7%	918	57.4%	1601	100.0%

statistical analysis: Pearson χ2 = 31.5 df = 3 *p* < 0.001.

## Data Availability

Not applicable.

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
