# Peer review of "Characteristics of Emergencies in the Workplace from the Perspective of the Emergency Medical Services: A 4-Year Case-Control Study"

_ijerph, 2023, doi:10.3390/ijerph20031863_

Round 1
Reviewer 1 Report
The paper is a retrospective study aiming to describe EMS calls to workplaces from 2015 to 2018 in Poland. While I think the manuscript focuses on an interesting an important topic, major revisions are required in order for this work to be acceptable for publication.
Introduction
While the aim of this paper is clearly stated by the authors at the end of the chapter, overall, I believe that the introduction could provide a more focused background of the situation in Poland. Indeed, the introduction succeeds in underlining the importance of providing adequate support to people of working age, it feels that more information regarding the statistics of work injuries/workplace fatalities. In fact, without this type of background, it is unclear why authors decided to focus their study on workplaces (is occupational mortality/morbidity a burden in Poland? Is workplace safety a concern? Are there many unsafe work environments?). Eg. in line 47 you mention the importance of health prevention programs, but it is unclear if they exist in Poland and to what extent. Some of this info are actually mentioned in the discussion but should be indeed introduced in this chapter. Additionally, readers would benefit from a more detailed description of what the SMR is.
Methods
Methods should be revised to improve clarity, in order to provide a clear overview of what was done and enough information to replicate the study. Additionally, this section contains many data that should actually be in the result section.
Detailed comments:
· Page 2 line 69 “The study area covered an area of 7,350 km2 and at the end of 2018 it was inhabited 69 by 547,866 people” what area are you referring to? Urban or rural? Could you provide a more precise description?
· Page 2 line 76 “Out of the analysed 155,993 EMS orders from 2015-2018, 1,725 cases (1.11%) were 76 defined as the place of the call indicated by the medical dispatcher or EMS manager 77 (1.11%), where there was no patient in 3 interventions, 1 was false, 1 was cancelled during 78 the arrival to the place of call, and in 111 the head of the EMS did not specify the patient’s 79 age.” This paragraph is unclear, I suggest rephrasing.
Results
This section features many tables and figures that could actually be merged into one “demographic” table providing info on the population in study (thus including total numbers, gender, codes…). There is no added value of having that many figures and table that do not really provide any novel result or important information. Readers should focus their attention on table 3,4 and 5 which contain the most interesting results, while instead this chapter is reach of excessive and redundant information.
Detailed comments:
· Page 5 line 147: “In turn, the 147 months of the most frequent interventions were: September (n=156; 9.74%) and November 148 (n=154; 9.62%) (Figure 3), days: Fridays (n=303; 18.93%) and Mondays (n=299; 18.68%), 149 and the hourly ranges: 08:01-12:00 (n=609; 38.04%) and 12:01-16:00 (n=494; 30.86%). EMS 150 teams were called to workplaces least frequently in winter (n=299; 18.68%), in the months 151 of: February (n=88; 5.50%) and January (n=113; 7.06%) (Figure 3), on: Sunday (n=45; 2.81%) 152 and Saturday (n=112; 7.00%), hours: 00:01-04:00 (n=52; 3.25%) and 20:01-24:00 (n=86; 153 5.37%). In the case of interventions for women, they most often took place in autumn 154 (n=210; 30.75%), September (n=74; 10.83%) and October (n=72; 10.54%) (Figure 3), on Fri-155 days (n=141; 20.64%) and Tuesdays (n=134; 19.62%), 08:01-12:00 (n=268; 39.24%) and 12:01-156 16:00 (n=227; 33.24%), and least frequently in winter (n=137; 20.06%), February (n=40; 157 5.86%) and July (n=44; 6.44), respectively %) (Figure 3), Sunday (n=19; 2.78%) and Satur-158 day (n=47; 6.88%), 00:01-04:00 (n=18; 2.64%) ) and 20:01-24:00 (n=54; 7.91%). In turn, EMS 159 team interventions in men’s workplaces were most common in the summer (n=261; 160 28.43%), in the months: June (n=92; 10.02%) and November (n=91; 9.91%) (Figure 3), Mon-161 day (n=179; 19.50%) and Thursday (n=165; 17.97%), hours: 08:01-12:00 (n=341; 37.15 %) 162 and 12:01-16:00 (n=267; 29.08%), the least frequent in winter (n=162; 17.65%), February 163 (n=48; 5.23%) and January (n=60; 6.51%) (Figure 3), Sunday (n=26; 2.81%) and Saturday 164 (n=65; 7.08%), 00:01-04:00 (n=34; 3.70%) and 8:01-24:00 (n=49; 5.34%).” This part is really unreadable. Readers should be guided by pointing out simple relationship and big-picture trends.
Discussion
The discussion (which is currently very long) could benefit a revision to better structure the flow and logic of the key findings in relation the available literature. I suggest to authors to better highlight key primary and secondary findings, stating if (and why) they are novel. Additionally, authors are presenting and summarizing data (eg page 9 line 272 “Our analyses show that EMS teams most often left for 272 work in autumn (28.54%), in September (9.74%), on Fridays (18.93%), between 08:01-12:00 273 (38 .04%), and least often in winter (18.68%), in February (5.50%), on Sundays (2.81%), 274 between 00:01-04:00 (n=49; 3, 25%)”, something that should have been done in the result section.
There is no real conclusion to the work, rather a list with a short recap of the main results.
Overall, the paper provides a description of the pattern of injuries reported in workplaces that required EMS intervention, failing to analyze and therefore explain to readers the implication of their results. For instance, it is unclear if the EMS calls to workplace represented a burden for the prehospital system (both in term of costs and in “pressuring” the system). Moreover, there is no information regarding in-hospital outcomes of this work-related injuries. Lack of information regarding the type of workplace leave no space to understand whether any direct health prevention program/work safety intervention is needed in some specific work sectors. In conclusion, this work does not currently provide any significant information that could contribute to advance the knowledge on this topic. I suggest the authors to investigate whether further analysis is possible with the data at their disposal.
Author Response
Introduction - While the aim of this paper is clearly stated by the authors at the end of the chapter, overall, I believe that the introduction could provide a more focused background of the situation in Poland. Indeed, the introduction succeeds in underlining the importance of providing adequate support to people of working age, it feels that more information regarding the statistics of work injuries/workplace fatalities. In fact, without this type of background, it is unclear why authors decided to focus their study on workplaces (is occupational mortality/morbidity a burden in Poland? Is workplace safety a concern? Are there many unsafe work environments?). Eg. in line 47 you mention the importance of health prevention programs, but it is unclear if they exist in Poland and to what extent. Some of this info are actually mentioned in the discussion but should be indeed introduced in this chapter. Additionally, readers would benefit from a more detailed description of what the SMR is.
Answer - Thank you for your valuable comments. The introduction was supplemented with information indicated by the reviewer. The problem of safety in the workplace was indicated both in terms of the European Union and Poland.
Methods - Methods should be revised to improve clarity, in order to provide a clear overview of what was done and enough information to replicate the study. Additionally, this section contains many data that should actually be in the result section. Detailed comments: Page 2 line 69 “The study area covered an area of 7,350 km2 and at the end of 2018 it was inhabited 69 by 547,866 people” what area are you referring to? Urban or rural? Could you provide a more precise description? Page 2 line 76 “Out of the analysed 155,993 EMS orders from 2015-2018, 1,725 cases (1.11%) were 76 defined as the place of the call indicated by the medical dispatcher or EMS manager 77 (1.11%), where there was no patient in 3 interventions, 1 was false, 1 was cancelled during 78 the arrival to the place of call, and in 111 the head of the EMS did not specify the patient’s 79 age.” This paragraph is unclear, I suggest rephrasing.
Answer - Thank you for your valuable comments. The methodology section was revised as recommended by the reviewer.
Results This section features many tables and figures that could actually be merged into one “demographic” table providing info on the population in study (thus including total numbers, gender, codes…). There is no added value of having that many figures and table that do not really provide any novel result or important information. Readers should focus their attention on table 3,4 and 5 which contain the most interesting results, while instead this chapter is reach of excessive and redundant information. Detailed comments: Page 5 line 147: “In turn, the 147 months of the most frequent interventions were: September (n=156; 9.74%) and November 148 (n=154; 9.62%) (Figure 3), days: Fridays (n=303; 18.93%) and Mondays (n=299; 18.68%), 149 and the hourly ranges: 08:01-12:00 (n=609; 38.04%) and 12:01-16:00 (n=494; 30.86%). EMS 150 teams were called to workplaces least frequently in winter (n=299; 18.68%), in the months 151 of: February (n=88; 5.50%) and January (n=113; 7.06%) (Figure 3), on: Sunday (n=45; 2.81%) 152 and Saturday (n=112; 7.00%), hours: 00:01-04:00 (n=52; 3.25%) and 20:01-24:00 (n=86; 153 5.37%). In the case of interventions for women, they most often took place in autumn 154 (n=210; 30.75%), September (n=74; 10.83%) and October (n=72; 10.54%) (Figure 3), on Fri-155 days (n=141; 20.64%) and Tuesdays (n=134; 19.62%), 08:01-12:00 (n=268; 39.24%) and 12:01-156 16:00 (n=227; 33.24%), and least frequently in winter (n=137; 20.06%), February (n=40; 157 5.86%) and July (n=44; 6.44), respectively %) (Figure 3), Sunday (n=19; 2.78%) and Satur-158 day (n=47; 6.88%), 00:01-04:00 (n=18; 2.64%) ) and 20:01-24:00 (n=54; 7.91%). In turn, EMS 159 team interventions in men’s workplaces were most common in the summer (n=261; 160 28.43%), in the months: June (n=92; 10.02%) and November (n=91; 9.91%) (Figure 3), Mon-161 day (n=179; 19.50%) and Thursday (n=165; 17.97%), hours: 08:01-12:00 (n=341; 37.15 %) 162 and 12:01-16:00 (n=267; 29.08%), the least frequent in winter (n=162; 17.65%), February 163 (n=48; 5.23%) and January (n=60; 6.51%) (Figure 3), Sunday (n=26; 2.81%) and Saturday 164 (n=65; 7.08%), 00:01-04:00 (n=34; 3.70%) and 8:01-24:00 (n=49; 5.34%).” This part is really unreadable. Readers should be guided by pointing out simple relationship and big-picture trends.
Answer - Thank you for drawing attention to this important aspect. The results section has been revised as recommended by the reviewer. Duplicate data has been removed.
Discussion - The discussion (which is currently very long) could benefit a revision to better structure the flow and logic of the key findings in relation the available literature. I suggest to authors to better highlight key primary and secondary findings, stating if (and why) they are novel. Additionally, authors are presenting and summarizing data (eg page 9 line 272 “Our analyses show that EMS teams most often left for 272 work in autumn (28.54%), in September (9.74%), on Fridays (18.93%), between 08:01-12:00 273 (38 .04%), and least often in winter (18.68%), in February (5.50%), on Sundays (2.81%), 274 between 00:01-04:00 (n=49; 3, 25%)”, something that should have been done in the result section.
Answer - The data has been included in the RESULTS chapter.
There is no real conclusion to the work, rather a list with a short recap of the main results.
Answer - CONCLUSIONS has been revised as recommended by the reviewer.
Reviewer 2 Report
I thank the authors for the time devoted to creating science. Thank you for inviting me to write a review.
In this retrospective observational study the authors made an attempt to characterize the population of patients suffering from accidents at work. They showed that these are mostly young middle-aged men. Most of the events incidence took place in autumn and in the morning hours. This is a group of patients who are relatively healthy in terms of internal medicine, which is why prompt provision of appropriate help, especially in the case of traumatic events, gives a chance to return to full fitness and continue working. This is important for social and economic reasons.
First, I have a few technical comments that will improve the readability of your manuscript:
1. I believe percentages should be rounded to significant figures. For example, the information "47.17%" does not add more than "47%" but significantly reduces the readability of the data.
2. Line 108-110:
I suggest simplifying the sentence: "The criterion of statistical significance was P<0.05". Any scientist can interpret this information without further explanation.
3. Figures should be below the description in the text. For example, first in line 116 is figure 1, and only in lines 1212-124 is the figure description and reference. Figure 1 should be placed below line 124. This remark is generally applicable (Fig. 2, 3….. and all)
4. In Figure 1 there is no description for the Y axis of the graph. By default, this is "N", but you should mark it on the chart
5. Line 123: The value of the test is presented (χ2=81.85806, p<0.0001). There is no need to provide such precise data here and all manuscript. It is enough (X2=81.9, p<0.001). If the value of P is anywhere >0.05, then it is enough to round to one decimal place, e.g. P=0.2
6. Line 357: Chapter 5 is not "results" but "conclusion"
7. Reference - edit the references according to the format recommended by the editors of the journal
8. The title of the paper states that it is a case-control study.
Case-control study (definition): An observational study that looks for an association between a given exposure and the occurrence of a specific endpoint by comparing the exposure (percentage exposed) in a group of subjects who experienced an endpoint with the exposure in an appropriately matched group of control subjects who had an endpoint final did not occur.
Does the conducted study meet the criteria of a case-control study?
Other remarks:
1. Table 3: What is evidenced by the statistical significance referred to in the lower part of the table (Pearson χ2=165.4186 df=10 p<0.0001)? Between which data were there significant differences? Between men and women? Between individual diagnoses according to ICD-10? I don't understand this statistical analysis. A similar remark applies to Table 4 and Table 5.
You should focus the most attention on the thorough improvement of the "discussion" chapter. In my opinion, the discussion is too long, and its form requires rethinking.
1. The information in lines 216-231 in my opinion is not a discussion, but a statement of facts. Statement of facts and justification for conducting the study are part of the introduction, not discussion.
2. In the first part of the discussion, the most important results from the conducted research should be discussed in 1-2 sentences. I suggest starting the discussion on line 232: "Based on our ....".
3. The research done is important. It provides valuable information characterizing quite a large part of EMS team interventions. It is important to notice as many practical aspects as possible in this discussion, and this should be the focus of the discussion. For example, line 255-259 shows: "Aftyka and Rudnicka-Drożak [15], based on data from the Provincial Emergency Services in Lublin (Poland), regarding all dispositions, regardless of the patient's age and
place of the event, indicated a higher percentage of EMS interventions in women (51.1%)
than in men (48.9%), while Pittet et al. found that about half of the EMS instructions concerned men (range from 48.5% to 50.2%)".
Questions arise: is the difference of 49 vs. The 51% in favor of women have any practical rationale? In addition to citing the results of research by other authors in the discussion, I would be happy to read 1-2 sentences of the commentary that explain the legitimacy of quoting such a work. If it has no practical effect, is it worth devoting so much space to it in the discussion?
4. Lines 270-285: Here, too, I would expect a discussion on the translation of the obtained information into clinical practice. If most trips to accidents at work were on Fridays, shouldn't more attention be paid to safety checks at workplaces on the last day of the week? Maybe the author suggests other suggestions, or has a different explanation for the obtained results?
5. Lines 296-302: I don't understand the relevance of presenting this information in the discussion. They do not refer to the results of the analysis presented by the authors. This information may be important in terms of possible prediction of the patient's potential burden of disease depending on the workplace, but this was not the subject of the study. Such a form of presenting data from work [19] would be good in an illustrative work.
6. Lines 304-308: again, this is just a representation of other authors' research. 1-2 sentences of comment are missing as it relates to the results obtained by the authors. Has the number of accidents increased/decreased compared to 2017? Has the profile of diseases underlying EMS interventions (according to ICD-10) changed over the years? What impact can this have on the practice/training process?
7. Line 351-355 (study limitations): “Finally, it should be stated that currently there areno studies on the intervention of emergency medical service teams in the workplace in patients of working age. This work is an attempt to fill this gap and is a contribution to further in-depth analyses on the above issues” - This sentence does not present any limitations of the study. This sentence justifies why this study was attempted. Such sentences are characteristic of the introduction to the manuscript.
The limitations of the study include (for example):
- study based on a register of qualitative data.
- the study does not verify pre-hospital diagnoses with hospital-confirmed diagnoses.
(… and other limitations typical of retrospective studies)
Summary, the discussion is typically of a specific nature:
1. A short summary of the results obtained (without repeating the numerical values)
2. Relation of own results to the results of other authors
3. Evaluation of the translation of the obtained results into clinical practice
4. Discussion of limitations.
In its current form, the discussion takes place alongside the results of the obtained research and takes on the character of a review paper. I suggest revising the discussion according to the comments presented, which should make the manuscript more attractive.
Author Response
- I believe percentages should be rounded to significant figures. For example, the information "47.17%" does not add more than "47%" but significantly reduces the readability of the data.
Answer - Thank you for your valuable attention. It was included in the manuscript.
- Line 108-110:
I suggest simplifying the sentence: "The criterion of statistical significance was P<0.05". Any scientist can interpret this information without further explanation.
Answer - Thank you for your valuable attention. It was included in the manuscript.
- Figures should be below the description in the text. For example, first in line 116 is figure 1, and only in lines 1212-124 is the figure description and reference. Figure 1 should be placed below line 124. This remark is generally applicable (Fig. 2, 3….. and all)
Answer - Thank you for your valuable attention. It was included in the manuscript.
- In Figure 1 there is no description for the Y axis of the graph. By default, this is "N", but you should mark it on the chart
Answer - Thank you for your valuable attention. It was included in the manuscript.
- Line 123: The value of the test is presented (χ2=81.85806, p<0.0001). There is no need to provide such precise data here and all manuscript. It is enough (X2=81.9, p<0.001). If the value of P is anywhere >0.05, then it is enough to round to one decimal place, e.g. P=0.2
Answer - Thank you for your valuable attention. It was included in the manuscript.
- Line 357: Chapter 5 is not "results" but "conclusion"
Answer - Thank you for your valuable attention. It was included in the manuscript.
- Reference - edit the references according to the format recommended by the editors of the journal
Answer - Thank you for your valuable attention. It was included in the manuscript. References have been supplemented.
- The title of the paper states that it is a case-control study.
Case-control study (definition): An observational study that looks for an association between a given exposure and the occurrence of a specific endpoint by comparing the exposure (percentage exposed) in a group of subjects who experienced an endpoint with the exposure in an appropriately matched group of control subjects who had an endpoint final did not occur.Does the conducted study meet the criteria of a case-control study?
Answer - Determination of the causes of EMS intervention (illness and accidents at the workplace) along with further management of the patient (on-site assistance, death, hospitalization, etc.) - according to the authors, it meets the requirements of a case-control study
Other remarks:
- Table 3: What is evidenced by the statistical significance referred to in the lower part of the table (Pearson χ2=165.4186 df=10 p<0.0001)? Between which data were there significant differences? Between men and women? Between individual diagnoses according to ICD-10? I don't understand this statistical analysis. A similar remark applies to Table 4 and Table 5.
Answer - The statistical analysis is explained in the table. The test examined the relationship between the patient's gender (female, male) and variables in the form of an urgency code, patient's age, ICD-10 group, diagnosis, type of action completion, etc.
You should focus the most attention on the thorough improvement of the "discussion" chapter. In my opinion, the discussion is too long, and its form requires rethinking.
- The information in lines 216-231 in my opinion is not a discussion, but a statement of facts. Statement of facts and justification for conducting the study are part of the introduction, not discussion.
Answer - Thank you for your valuable attention. It was included in the discussion.
- In the first part of the discussion, the most important results from the conducted research should be discussed in 1-2 sentences. I suggest starting the discussion on line 232: "Based on our ....".
Answer - Thank you for your valuable attention. It was included in the discussion.
- The research done is important. It provides valuable information characterizing quite a large part of EMS team interventions. It is important to notice as many practical aspects as possible in this discussion, and this should be the focus of the discussion. For example, line 255-259 shows: "Aftyka and Rudnicka-Drożak [15], based on data from the Provincial Emergency Services in Lublin (Poland), regarding all dispositions, regardless of the patient's age and place of the event, indicated a higher percentage of EMS interventions in women (51.1%) than in men (48.9%), while Pittet et al. found that about half of the EMS instructions concerned men (range from 48.5% to 50.2%)". Questions arise: is the difference of 49 vs. The 51% in favor of women have any practical rationale? In addition to citing the results of research by other authors in the discussion, I would be happy to read 1-2 sentences of the commentary that explain the legitimacy of quoting such a work. If it has no practical effect, is it worth devoting so much space to it in the discussion?
Answer - The literature on the main subject of the article is quite poor, which is why the authors cite the available literature, comparing it with the results obtained. Following the recommendation of the reviewer, the discussion was limited to the most important issues.
- Lines 270-285: Here, too, I would expect a discussion on the translation of the obtained information into clinical practice. If most trips to accidents at work were on Fridays, shouldn't more attention be paid to safety checks at workplaces on the last day of the week? Maybe the author suggests other suggestions, or has a different explanation for the obtained results?
Answer - Thank you for your valuable attention. It was included in the discussion.
- Lines 296-302: I don't understand the relevance of presenting this information in the discussion. They do not refer to the results of the analysis presented by the authors. This information may be important in terms of possible prediction of the patient's potential burden of disease depending on the workplace, but this was not the subject of the study. Such a form of presenting data from work [19] would be good in an illustrative work.
Answer - Thank you for your valuable attention. The fragment has been removed.
- Lines 304-308: again, this is just a representation of other authors' research. 1-2 sentences of comment are missing as it relates to the results obtained by the authors. Has the number of accidents increased/decreased compared to 2017? Has the profile of diseases underlying EMS interventions (according to ICD-10) changed over the years? What impact can this have on the practice/training process?
Answer - Thank you for your valuable attention. Data on the number of EMT interventions in individual years were supplemented, and the Discussion and Conclusions were supplemented based on this data. The number of interventions was: 2015 n=281; 2016 n=356; 2017 n=448; 2018 n=516. This has been included in the results.
- Line 351-355 (study limitations): “Finally, it should be stated that currently there areno studies on the intervention of emergency medical service teams in the workplace in patients of working age. This work is an attempt to fill this gap and is a contribution to further in-depth analyses on the above issues” - This sentence does not present any limitations of the study. This sentence justifies why this study was attempted. Such sentences are characteristic of the introduction to the manuscript.
Answer - Thank you for your valuable attention. The fragment has been removed. The limitations of the study have been supplemented again.
Round 2
Reviewer 2 Report
Accept in present form.